# Magnetic Nanoparticles Enhanced Surface Plasmon Resonance Biosensor for Rapid Detection of *Salmonella* Typhimurium in Romaine Lettuce

**DOI:** 10.3390/s22020475

**Published:** 2022-01-09

**Authors:** Devendra Bhandari, Fur-Chi Chen, Roger C. Bridgman

**Affiliations:** 1Department of Agricultural and Environmental Sciences, Tennessee State University, Nashville, TN 37209, USA; dbhandar@my.tnstate.edu; 2Department of Human Sciences, Tennessee State University, Nashville, TN 37209, USA; 3Hybridoma Facility, Auburn University, Auburn, AL 36830, USA; bridgcr@auburn.edu

**Keywords:** biosensor, *Salmonella* Typhimurium, surface plasmon resonance, magnetic nanoparticle, flagellin, monoclonal antibody

## Abstract

*Salmonella* is one of the major foodborne pathogens responsible for many cases of illnesses, hospitalizations and deaths worldwide. Although different methods are available to timely detect *Salmonella* in foods, surface plasmon resonance (SPR) has the benefit of real-time detection with a high sensitivity and specificity. The purpose of this study was to develop an SPR method in conjunction with magnetic nanoparticles (MNPs) for the rapid detection of *Salmonella* Typhimurium. The assay utilizes a pair of well-characterized, flagellin-specific monoclonal antibodies; one is immobilized on the sensor surface and the other is coupled to the MNPs. Samples of romaine lettuce contaminated with *Salmonella* Typhimurium were washed with deionized water, and bacterial cells were captured on a filter membrane by vacuum filtration. SPR assays were compared in three different formats—direct assay, sequential two-step sandwich assay, and preincubation one-step sandwich assay. The interaction of flagellin and MNPs with the antibody-immobilized sensor surface were analyzed. SPR signals from a sequential two-step sandwich assay and preincubation one-step sandwich assay were 7.5 times and 14.0 times higher than the direct assay. The detection limits of the assay were 4.7 log cfu/mL in the buffer and 5.2 log cfu/g in romaine lettuce samples.

## 1. Introduction

In recent years, there has been a substantial increase in illnesses caused by consuming foods contaminated with pathogenic microorganisms. Of all the foodborne pathogens, non-typhoidal *Salmonella* is one of the major pathogens that is a risk to the public. It is estimated that, in United States, *Salmonella* is responsible for 1,027,561 cases of illnesses, 19,336 hospitalizations and 378 deaths annually [1], costing USD 3.3 billion due to associated medical costs, productivity loss and premature mortality [2]. Worldwide, *Salmonella enterica* serovar Typhimurium (*S.* Typhimurium) is the second most common serotype (after *Salmonella enterica* serovar Enteritidis) that causes foodborne illnesses [3]. Therefore, to timely address public health issues and avoid the huge cost associated with illnesses caused by *S.* Typhimurium, it is necessary to develop rapid, sensitive and specific detection methods.

Various methods were developed for the isolation and identification of *S.* Typhimurium in food samples. Culture-based methods are laborious and time-consuming, taking 3–4 days for preliminary identification and 5–7 days for confirmation [4]. As a result, various methods, such as immunoassays and molecular methods, were developed for the detection of *S.* Typhimurium and other foodborne pathogens [5]. Although immunological methods such as the enzyme-linked immunosorbent assay (ELISA) are faster than culture-based methods [6] and were used for the detection of *S.* Typhimurium [7,8,9], they require a time-consuming enrichment and sample pre-treatment steps. In recent years, PCR methods were reported as more specific, rapid and sensitive, and were extensively used for detection of *Salmonella* [10,11,12,13,14]. However, major drawbacks of PCR methods, including difficulty in automation, requirement of sample pre-enrichment [11,15,16] and amplification inhibition [17,18,19] by food components and chemicals present in the selective enrichment media have left avenues to develop more efficient and reliable detection methods.

Recently, the surface plasmon resonance (SPR) biosensor has gained attention in the detection of foodborne pathogens because it allows the fast, label-free and real-time monitoring of the biomolecular interactions with a higher sensitivity and specificity [20]. SPR biosensors measure the changes in the refractive index caused by changes in mass, which result from analyte and ligand binding at the metal interface [21]. Various studies reported the use of SPR biosensors for the detection of *Salmonella*, with an improved specificity and sensitivity to some extent [22,23,24,25,26,27]. However, most of these earlier SPR detection methods required sample enrichment steps, which increased the detection time. Additionally, most of the SPR detection methods were not developed for solid food samples such as fruits and vegetables.

The sensitivity of SPR can be enhanced by using dense particles conjugated with an antibody. It is reported that magnetic nanoparticles (MNPs) increase the SPR responses when used as signal amplifiers [28,29,30,31,32]. At the same time, MNPs provide the simultaneous purification of samples and reduce the background binding interferences. MNPs gained a special attention because of their higher surface-to-volume ratio, minimum disturbance to attached biomolecules, faster binding rates, higher miscibility, higher specificity [21] and easier separation using a magnetic field.

In this study, we report the development of a rapid and sensitive *S.* Typhimurium detection method using a SPR biosensor combined with the amplification of antibody-coupled MNPs. A monoclonal antibody specific to *S*. Typhimurium flagellin was coupled to superparamagnetic nanoparticles (50 nm) composed of iron oxide and polysaccharide (MACSflex MicroBead). *S.* Typhimurium cells were recovered by washing contaminated romaine lettuce followed by vacuum filtration. Flagellin was extracted using glycine-HCl (250 mM, pH 2.0). Three different assay formats, direct assay, sequential two-step sandwich assay and preincubation one-step sandwich assay, were compared for their sensitivity. Finally, a functional protocol to detect *S.* Typhimurium starting from romaine lettuce sample preparation to SPR detection was developed.

## 2. Materials and Methods

### 2.1. Materials and Instrument

*S*. Typhimurium (ATCC 13311) was acquired from American Type Culture Collection (Manassas, VA, USA) and stored at −80 °C before use. Packages of romaine lettuce were purchased from a local grocery store days before the experiments. Xylose-lysine-tergitol 4 (XLT4 agar, Remel, Thermo Fisher Scientific, Lenexa, KS, USA), plate count agar (PCA, Difco), tryptic soy agar (TSA, Remel), buffered peptone water (BPW, Remel), 3-((3-cholamidopropyl) dimethylammonio)-1-propanesulfonate (CHAPS), 10X phosphate buffered saline (10X PBS), TWEEN 20, and bovine serum albumin (BSA) were supplied by Thermo Fisher Scientific Inc. (Waltham, MA, USA); 1X PBS with 0.05% TWEEN 20 (PBST) was used as the working buffer.

Deionized water was purified with a Simplicity Water Purification System (MilliporeSigma, Burlington, MA, USA), and then degassed with a vacuum chamber. Deionized and degassed water was used to prepare all the reagents and solutions. PVDF membrane filters (0.45 μm, HVLP04700) were obtained from MilliporeSigma (Burlington, MA, USA). Fisherbrand glass microanalysis vacuum holders, HulaMixer (Invitrogen, Thermo Fisher Scientific, Lenexa, KS, USA), NanoDrop Lite Spectrophotometer, and Amicon Ultra-15, 10K Centrifugal Filter (MilliporeSigma) were purchased from Thermo Fisher Scientific (Waltham, MA, USA).

MACSflex MicroBead kit, µ-Columns, MACS MultiStand and µMACS Separator were purchased from Miltenyi Biotec (Bergisch Gladbach, Germany), and 1X PBS with 1% Triton X-100 was used as the equilibration buffer to wash and elute antibody-coupled MNPs–flagellin complexes. *N*-(3-Dimethylaminopropyl)-N′-ethylcarbodiimide hydrochloride (EDC), *N*-Hydroxysuccinimide (NHS), ethanolamine hydrochloride, sodium acetate, sodium chloride, glycine and triton X-100 were purchased from Sigma-Aldrich Inc. (St. Louis, MO, USA).

Reichert duel-channel SR7500DC biosensor (SPR biosensor) and its associated software, called Integrated SPRAutolink Version 1.1.14-T from Reichert Technologies (Buffalo, NY, USA), were utilized to perform experiments and quantify the binding of analyte with the immobilized antibody. TraceDrawer Version 1.6.1 by Ridgeview Instruments AB (Upsala, Sweden) was used to further process and analyze SPR data. The 500 kDa Carboxymethyl Dextran Hydrogel Surface Sensor Chip (SR7000 gold sensor slide) was purchased from Reichert Inc. (Buffalo, NY, USA). Monoclonal antibodies (MAb 1E10 and MAb 1C8) specific to *S.* Typhimurium flagellin were produced in our laboratories, as described in a previous study [33].

### 2.2. Preparation of Antibody-Coupled Magnetic Nanoparticles

MAb 1C8 was coupled to magnetic nanoparticles (MNPs) at the rate of 700 pmole/mg of MACSflex MicroBead. MAb 1C8 (106.5 µg) was mixed with reconstitution buffer (450 µL), and the mixture was used to rehydrate the lyophilized MACSflex MicroBead (0.5 mg). After 2 h of incubation at room temperature in dark place, the mixture was further incubated for 24 h at 4 °C. The next day, MAb 1C8-coupled MNPs were captured and eluted in the storage buffer provided in the kit using a µ-Column and a µMACS Separator following manufacturer’s instructions. Final concentrations of antibody-coupled MNPs were adjusted to 1.25 mg/mL using the storage buffer. The prepared antibody-coupled MNPs were stored at 4 °C and used within 4 weeks.

### 2.3. Preparation of Flagellin Extraction

Cultures of *S.* Typhimurium were prepared on TSA agar plates and incubated at 35 °C for 24 h after retrieving from the freezer. The cultures were further propagated in buffered peptone water (BPW) by incubating overnight at 37 °C. Concentration of bacteria in BPW was determined by enumeration on PCA agar plates. The aliquots of BPW (1 mL) were centrifuged at 3000× *g* for 10 min and the cell pellets were collected. Cell pellets were suspended in 100 µL glycine-HCl (250 mM, pH 2.0) and incubated for 30 min at room temperature to extract flagellin. The volume of suspension was adjusted to 500 µL by adding 50 µL of 10X PBST and 350 µL of degassed–deionized water. The pH of the suspension was adjusted to 7.0 and then centrifuged at 3000× *g* for 10 min. The supernatants were collected, and the protein concentrations were measured using a NanoDrop Lite spectrophotometer. The flagellin preparations were analyzed using Bolt 4–12% Bis-Tris Plus gels to check for purity. Presence of minor fragments in the flagellin samples was noticed as described in our previous study [33]. The flagellin samples were stored at −80 °C.

### 2.4. Preparation of SPR Sensor Surface

SPR sensor chip preparation and immobilization of monoclonal antibody were performed as described in earlier publications [27,33]. The 500 kDa carboxymethyl dextran hydrogel surface sensor chip (SR7000 gold sensor slide) was installed onto Reichert SR7500DC biosensor following the manufacturer’s instruction. The sensor surface was then pre-conditioned by running PBST at 20 µL/min until a stable baseline was obtained. The flow rate of 20 µL/min and temperature of 25 °C were maintained throughout the immobilization process. In order to activate carboxyl groups on the surface of the sensor chip, fresh preparation of 40 mg EDC and 10 mg NHS dissolved in 1 mL of deionized water was injected onto the sensor surface for 5 min. After surface activation, MAb 1E10 diluted in 10 mM sodium acetate (150 µg/mL, pH 5.2) was injected to the left channel of the surface for 5 min. Then, BSA dissolved in 10 mM sodium acetate (75 µg/mL, pH 5.2) was injected to both channels to saturate the remaining active sites. Finally, quenching solution (1.0 M ethanolamine, pH 8.5) was injected for 5 min to deactivate the carboxyl groups and wash away the unbound antibody and BSA. A continuous flow of running buffer (PBST) at 20 µL/min was maintained after the completion of antibody immobilization. SPR assays were carried out after a stable baseline was achieved. All experiments were performed at a constant temperature of 25 °C. Filtered and degassed PBST was used as the running buffer at 20 µL/min.

### 2.5. SPR Assay Formats

Three SPR assay formats were compared in this study, as illustrated in Figure 1. The first was a direct assay in which flagellin sample was directly injected onto the SPR sensor surface immobilized with MAb 1E10 (Figure 1a). In this format, the SPR response was proportionately related to the flagellin captured by the MAb 1E10 on the sensor surface. The second was a sequential two-step sandwich assay in which the antibody-coupled MNPs (41.6 µg/mL) were injected after the direct assay to form a sandwich on the captured flagellin (Figure 1b). The antibody-coupled MNPs provide an amplification to the SPR response of the captured flagellin in the direct assay. In this format, two responses were recorded, one from the direct assay and another from the binding of antibody-coupled MNPs to the captured flagellin. The third was a preincubation one-step sandwich assay in which the antibody-coupled MNPs (41.6 µg/mL MNPs) were incubated (30 min, room temperature) with the flagellin sample before injection (Figure 1c). After incubation, flagellin-antibody-coupled MNPs complexes were separated using a µ-Column and MACSflex separator. The µ-Column was first equilibrated with 1X PBS supplemented with 1% Trition-100, and the flagellin-antibody-coupled MNPs complexes captured by the µ-Column were washed with PBST. Finally, the complexes were eluted in 300 µL of PBST, and then injected onto the sensor surface. In this format, the flagellin-antibody-coupled MNPs complexes formed during the incubation contributed to a more augmented SPR response than the flagellin alone when captured by MAb 1E10 on the sensor surface.

### 2.6. Preparation of Flagellin Sample from Romaine Lettuce

Twenty-five grams of romaine lettuce samples were inoculated with different levels of *S*. Typhimurium (5.5, 6.0 and 6.5 log cfu/g) and washed gently with 100 mL of deionized water. Romaine lettuce samples were then discarded, and the wash solutions were filtered under vacuum. Filter membrane with trapped bacteria was placed in a large centrifuge tube (50 mL). A volume of 10 mL glycine-HCl (250 mM, pH 2.0) was added to the tube and vortexed at high speed for 2 min followed by a 30 min incubation at room temperature with constant mixing on a HulaMixer. Filter membrane was discarded after incubation, and pH of the solution was adjusted to 7.0. The solution was centrifuged (7000× *g*, 20 min), and the supernatant was recovered and concentrated using an Amicon Ultra-15 centrifugal filter (7000× *g*, 20 min). Volumes of 10 mL PBST were added twice to the centrifugal filter to change buffer to PBST and centrifugation continued until the volume left in the filter was about 100 µL. The concentrated flagellin sample was collected in a microcentrifuge tube and centrifuged at 16,000× *g* for 5 min, the supernatant was collected, and the volume was adjusted to 100 µL with PBST.

Antibody-coupled MNPs (41.6 µg/mL) were mixed with the concentrated flagellin sample, and then incubated for 30 min at room temperature with constant mixing on a HulaMixer. After incubation, flagellin-antibody-coupled MNPs complexes were separated using a µ-Column and MACSflex separator. The µ-Column was first equilibrated with 1X PBS supplemented with 1% Trition-100, and the flagellin-antibody-coupled MNPs complexes captured by the µ-Column were washed with PBST. Finally, the complexes were eluted in 300 µL of PBST, and then injected onto the sensor surface. A summary of the preparation of the flagellin sample from romaine lettuce for the SPR assay is presented in Table 1. The processing time needed to complete the entire procedure was generally less than 4 h.

## 3. Results

### 3.1. Optimization of Antibody-Coupled Magnetic Nanoparticles

To determine the performance and cost effectiveness of antibody-coupled MNPs for the amplification of the SPR response, different concentrations of antibody-coupled MNPs were individually tested using the sequential two-step sandwich assay. As shown in Figure 2a, the SPR responses were dependent on the concentrations of antibody-coupled MNPs. It was evident that the amplified SPR responses were directly proportional to the concentrations of antibody-coupled MNPs (Figure 2b). On average, the SPR response to flagellin (9.2 µg/mL) alone was 43.2 ± 2.9 µRIU in the direct assay. For the subsequent amplifications using different concentrations of antibody-coupled MNPs (10.4, 20.8, 41.6, and 83.2 µg/mL), the SPR responses were significantly increased at all concentrations. The ratio of the SPR response to the concentration of antibody-coupled MNPs is 7.2 µRIU/µg/mL. A higher amplification was achieved when a higher concentration of antibody-coupled MNPs was used; however, the cost of the assay would be higher. Based on the results, the concentration of 41.6 µg/mL antibody-coupled MNPs was selected and applied to the subsequent experiments as this level is economically feasible and provides sufficient amplifications.

### 3.2. Comparison of Three SPR Assay Formats

Three SPR formats (direct assay, sequential two-step sandwich assay and preincubation one-step sandwich assay) were performed to compare the SPR signal amplification. The same concentrations of flagellin extractions (9.2 µg/mL) from *S*. Typhimurium were analyzed with the same concentrations of MNPs (41.6 µg/mL) used in both the sequential two-step sandwich assay and preincubation one-step sandwich assay. The results showed that the SPR signals from the direct assay, sequential two-step sandwich assay and preincubation one-step sandwich assay were 43.2 ± 2.9 µRIU, 326.5 ± 14.4 µRIU and 602.9 ± 28.3 µRIU, respectively (Figure 3). The two sandwich assays have significantly amplified the detection signals. SPR responses from the sequential two-step sandwich assay and the preincubation one-step sandwich assay were 7.5 times and 14.0 times higher than the direct assay. It was evident that the preincubation one-step sandwich assay generated the highest detection signal, which was 1.9 times higher than the signal produced by the sequential two-step sandwich assay. Therefore, the preincubation one-step sandwich assay was further evaluated for the detection of *S.* Typhimurium in the following experiments.

### 3.3. Detection of S. Typhimurium from Enriched BPW

The preincubation one-step sandwich assay was applied for the detection of *S.* Typhimurium cultured in BPW. Samples of BPW-enriched *S*. Typhimurium were diluted to different concentrations (A = 5.0 log cfu/mL, B = 5.2 log cfu/mL, C = 5.6 log cfu/mL and D = 5.9 log cfu/mL), and flagellin extractions of the samples were prepared and analyzed. The SPR responses were 18.8 ± 6.3, 22.1 ± 5.8, 40.2 ± 5.3, and 119.6 ± 9.7 µRIU for A, B, C, and D, respectively (Figure 4). An exponential correlation between SPR responses and the concentrations of *S*. Typhimurium was observed within the experimental range, and thus was tailing at low concentrations. The detection limit was 4.7 log cfu/mL as determined from the deviations of baseline signals. The results suggested that the preincubation one-step sandwich assay was sensitive enough for the detection of *S*. Typhimurium from BPW enriched samples in a standard overnight enrichment protocol.

### 3.4. Detection of S. Typhimurium from Romaine Lettuce

A protocol was designed for the detection of *S*. Typhimurium directly from samples of romaine lettuce. *Salmonella* cells in the contaminated romaine lettuce were captured, and flagellin was extracted and concentrated using this protocol. As summarized in Table 1, the protocol consists of capturing *Salmonella* cells by a vacuum filtration, extraction of flagellin by glycine-HCl, concentration of flagellin by ultra-centrifugal filtration and detection of flagellin by the preincubation one-step sandwich assay. The total time of analysis required by the protocol was 200 min.

As this method does not include an overnight enrichment step, the recovery of bacterial cells directly from romaine lettuce samples is a critical aspect to the subsequent detection. To determine the recovery efficiency of washing and vacuum filtration procedures, four lettuce samples (25 g) inoculated with 7.4 log cfu of *S*. Typhimurium were rinsed with 100 mL of deionized water and the washing liquid was vacuumed through a filter membrane (0.45 µm). The number of *S.* Typhimurium in the samples before and after vacuum filtration was determined and the percentage of recovery was calculated. It was found that the average recovery of *S.* Typhimurium was 45% (ranging from 40–47%). This recovery percentage is not unexpected as some bacteria were not removed from the samples; therefore, it is important to consider the recovery percentage when interpreting the data of *Salmonella* contamination levels in the samples in a quantitative manner.

To further evaluate the performance of this analytical protocol, samples of romaine lettuce were artificially contaminated with *S*. Typhimurium at different levels (5.5, 6.0 and 6.5 log cfu/g) and analyzed following the same protocol. The results indicated that *S*. Typhimurium in romaine lettuce samples can be detected with this protocol at all three levels. The SPR responses obtained from the romaine lettuce inoculated at 5.5, 6.0 and 6.5 log cfu of *S*. Typhimurium per gram of sample were 24.1 ± 6.0, 92.8 ± 21.2, and 162.1 ± 33.0 µRIU, respectively (Figure 5). This protocol significantly reduced the overall detection time as it did not require overnight enrichment. The analytical process from the sample preparation to SPR detection can be completed within 4 h. The detection limit of *S*. Typhimurium in the romaine lettuce sample was determined as 5.2 log cfu/g based on the deviations of the baseline signals.

## 4. Discussion

We demonstrated the amplification of SPR signals using antibody-coupled MNPs for the detection of *S.* Typhimurium. The antibody-coupled MNPs served as multifunctional mediators for flagellin capturing, purification, concentration, and signal amplification. The highest amplification was achieved in the preincubation one-step sandwich assay when the antibody-coupled MNPs were incubated with flagellin extractions. There were 14-fold increases in the SPR responses from the direct assay and almost 2-fold increases in the sequential two-step sandwich assay. Comparisons of the performances of the three SPR assays are provided in Table 2. This nature of amplification in SPR signals was reported earlier in another study which observed significant increases in detection signals when Staphylococcal enterotoxin B (SEB) incubated with anti-SEB, antibody-conjugated MNPs were used for the SPR detection [34]. In our study, what ultimately amplified the SPR signal is the complex of flagellin-antibody-coupled MNPs formed as the result of preincubation. This complex consists of flagellin, IgG antibody and MNP, which lead to a net increase in mass of the analyte and a higher change of reflective index on the sensor surface [21].

We also assert that SPR in combination with MNPs significantly increased the detection sensitivity. In contrast to other *Salmonella* detection methods, this SPR assay does not require an enrichment procedure. In the buffer system, the detection limit of the assay was 4.7 log cfu/mL, and in the romaine lettuce sample, the detection limit of the assay was 5.2 log cfu/g. The loss of detection sensitivity in the romaine lettuce sample was mostly due to the lower recovery and matrix effects from the sample. There were some difficulties in performing the vacuum filtration of washing liquid from romaine lettuce samples because of presence of the debris of leaf tissues and particles from soil. These substances may block the pores of the filter membrane during vacuum filtration, thus demanding longer processing time. This problem can be addressed either by changing the filter membrane or by using a stirring device with a vacuum filter.

In this study, the recovery of *S.* Typhimurium from the romaine lettuce sample after washing and vacuum filtration was less than 50%. This lower recovery could ultimately contribute to the lower detection signal and lower sensitivity. This low recovery is largely caused by the uneven surface of the romaine lettuce where bacteria can be easily trapped. There were significant variations in the efficiency of recovery, as found in the literature. The removal of bacteria from the surfaces was dependent on the types of food sample and the methods of recovery. For instance, one study reported that 55% of *Escherichia coli* were captured using antibody-coupled magnetic nanoparticles [35]. On the other hand, another study reported that the recovery of *Salmonella enteritidis* from eggshell was more than 90% using antibody-coupled magnetic nanoparticles [32]. It is anticipated that the recovery of *S.* Typhimurium from food surfaces can be further improved by designing more effective and food-matrix-specific strategies to ultimately increase the detection sensitivity.

It is crucial for any detection method to have capability of detecting lower levels of *Salmonella* contamination in a food system. Therefore, the enrichment is a prerequisite for the detection of *Salmonella* in cultural methods [4], PCR methods [11,15,16] and conventional SPR methods [23,24,25,27]. If a low level of contamination is expected, an enrichment procedure would be needed in conjunction with the current protocol. We showed that low levels of S. Typhimurium contamination (less than 1.0 log cfu/g) in leafy vegetables can be successfully detected by the SPR assays following BPW enrichment [27].

It is important to note that fresh vegetables can harbor large and diverse populations of bacteria. Therefore, the specificity of an SPR assay is even more important than the sensitivity and should be demonstrated when validating the SPR assay. Monoclonal antibodies used in this study are specific to the flagellin (phase 1 and phase 2) of *S*. Typhimurium and do not recognize ligands other than flagellin. Additionally, these monoclonal antibodies are well characterized and recognize distinct epitopes on the flagellin of *S*. Typhimurium [33]. The specificity of the SPR assay using these monoclonal antibodies was further confirmed with commensal bacteria isolated from the romaine lettuce samples [27].

In conclusion, we demonstrated that the application of MNPs with flagellin-specific monoclonal antibodies in the detection of *S*. Typhimurium is an effective way to amplify the SPR response. By comparing different SPR assay formats, we concluded that the preincubation one-step sandwich assay provides the most enhancement to the SPR signal and has the ability to detect a lower number of *Salmonella* than the direct assay and sequential two-step sandwich assay. In addition, we showed that *S*. Typhimurium contamination in leafy vegetables can be successfully detected without overnight enrichment. SPR assays for other serotypes of *Salmonella* can be developed on the same ground of this work using serotype-specific antibodies. Finally, our results demonstrate the potential of an SPR biosensor in combination with MNPs for the detection of other pathogens and biomolecules with a higher sensitivity and specificity.

## Figures and Tables

**Figure 1 sensors-22-00475-f001:**
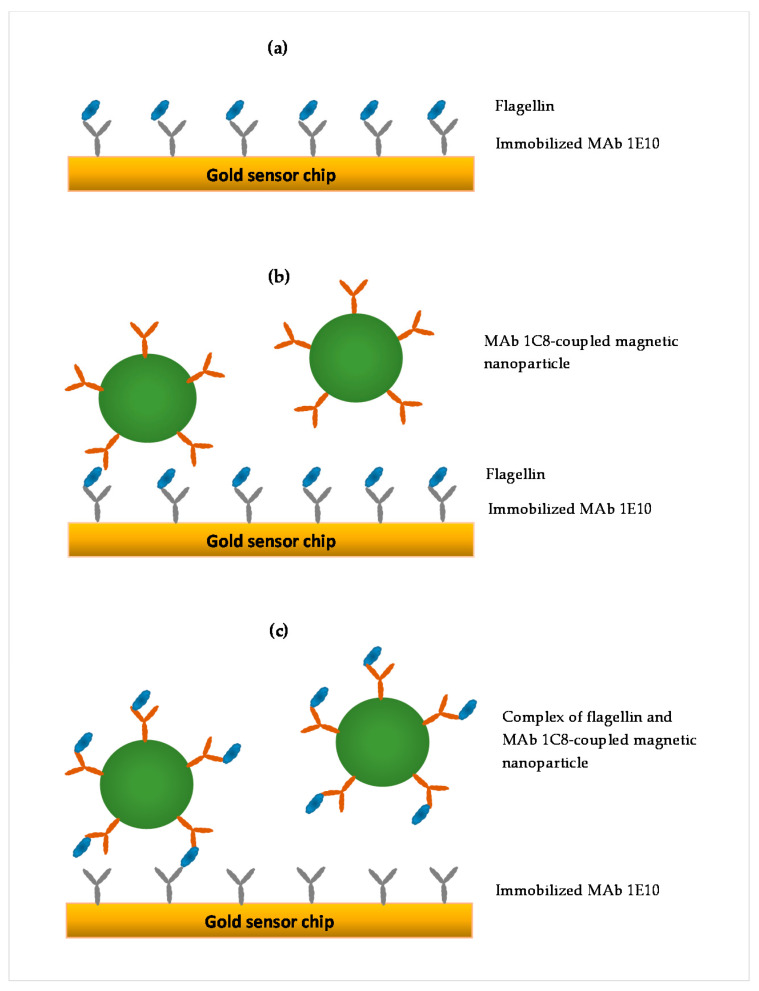
Illustrations of three different SPR assay formats: (**a**) direct assay—sample preparation is injected directly, (**b**) sequential two-step sandwich assay—MAb 1C8-coupled magnetic nanoparticle is injected following sample injection, and (**c**) preincubation one-step sandwich assay—sample preparation is preincubated with MAb 1C8-coupled magnetic nanoparticle before injection.

**Figure 2 sensors-22-00475-f002:**
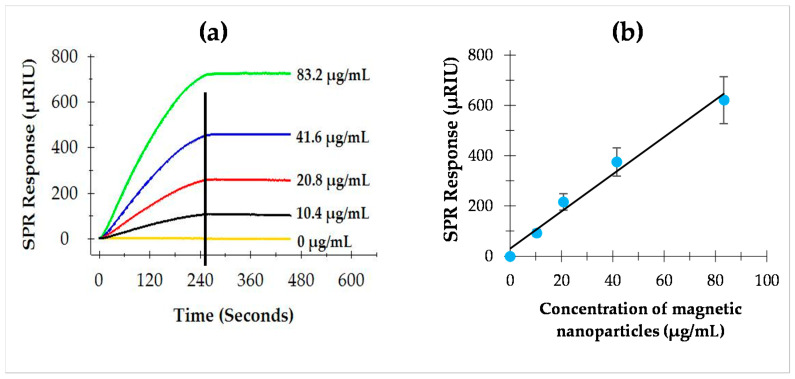
(**a**) SPR sensorgrams of the sequential two-step sandwich assay showing the interaction of MAb 1C8-coupled magnetic nanoparticles at various concentrations when the flagellin is captured on the MAb 1E10-immobilized sensor surface: (**b**) SPR responses at the peak of association phase, the vertical line in (**a**), showing a linear relation to the concentrations of MAb 1C8-coupled magnetic nanoparticles.

**Figure 3 sensors-22-00475-f003:**
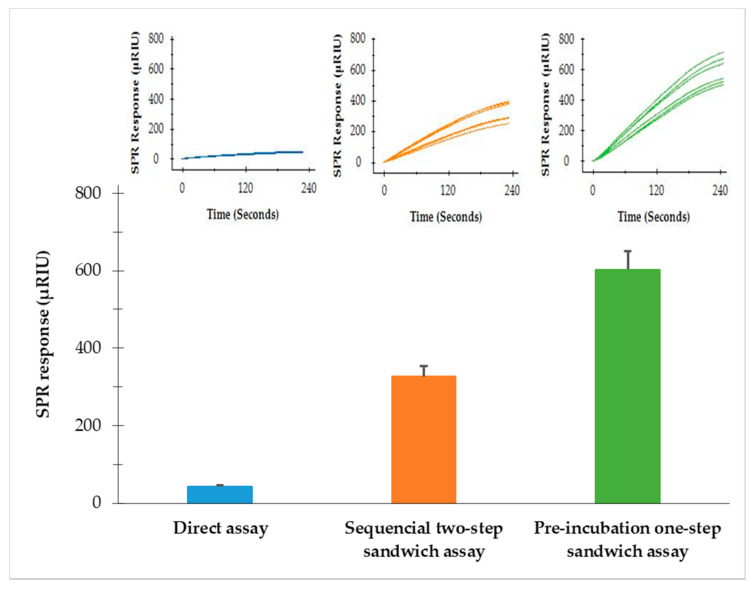
SPR responses at the peak of association phase of the three assay formats—direct assay, sequential two-step sandwich assay and preincubation one-step sandwich assay; same concentrations of flagellin extractions (9.2 µg/mL) from *S*. Typhmurium were analyzed. Same concentration of MNPs (41.6 µg/mL) was used in the sequential two-step sandwich assay and preincubation one-step sandwich assay. Two experiments were performed on different days, each with three replicates.

**Figure 4 sensors-22-00475-f004:**
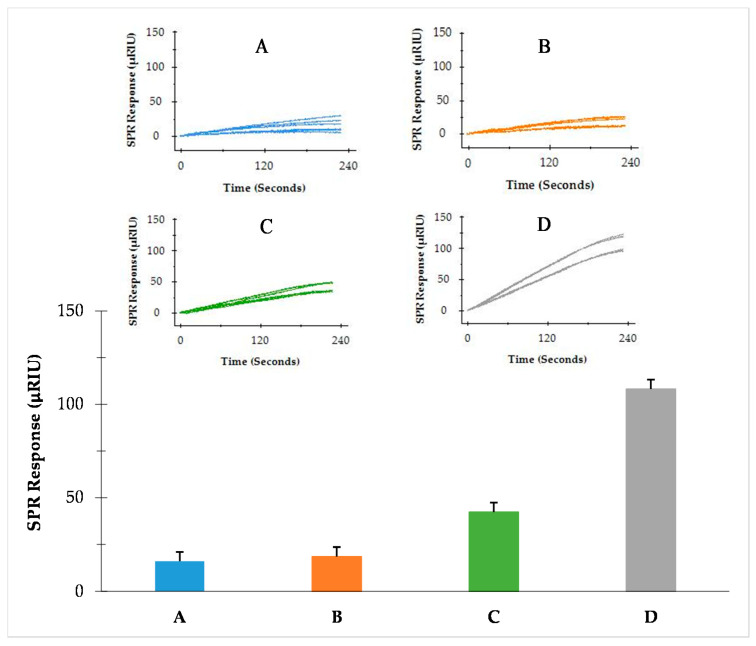
SPR responses of *S*. Typhimurium at various concentrations (**A**) 5.0 log cfu/mL, (**B**) 5.2 log cfu/mL, (**C**) 5.6 log cfu/mL and (**D**) 5.9 log cfu/mL, analyzed by the preincubation one-step sandwich assay. Two experiments were performed in different days each with three replicates.

**Figure 5 sensors-22-00475-f005:**
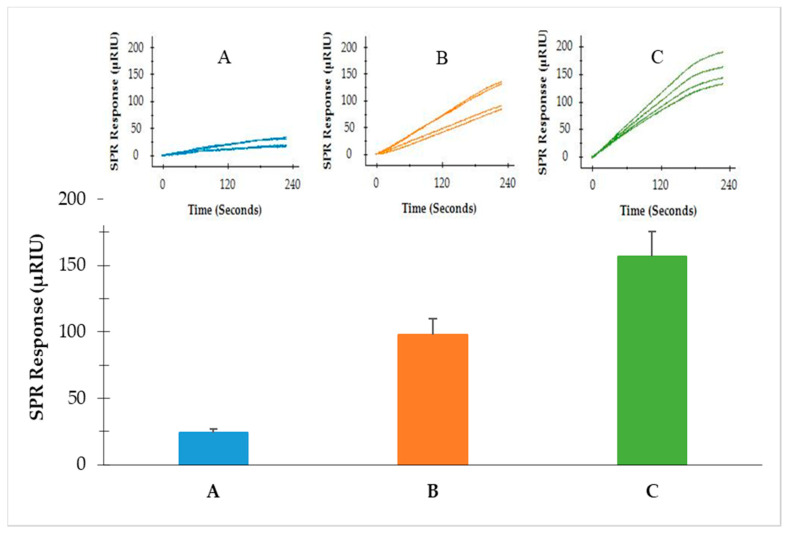
SPR responses of the romaine lettuce samples contaminated with different levels of *S*. Typhimurium (**A**) 5.5 log cfu/g, (**B**) 6.0 log cfu/g and (**C**) 6.5 log cfu/g, analyzed by the preincubation one-step sandwich assay. Two experiments were performed on different days, each with two replicates.

**Table 1 sensors-22-00475-t001:** A summary of the preparation of flagellin sample from romaine lettuce for the SPR assay.

Steps	Procedures	Minutes
1	Romaine lettuce is washed with deionized water, and the washing liquid is removed and filtered under vacuum to collect solid matters.	20
2	Filter membrane is submerged in glycine-HCl (250 mM, pH 2.0) solution and held on a shaker.	30
3	Solution of pH is adjusted to 7 and supernatant is collected by centrifugation.	20
4	Supernatant is concentrated using an Amicon centrifugal filter.	20
5	Supernatant is exchanged into PBST buffer using an Amicon centrifugal filter.	40
6	Supernatant is collected, and the final volume is adjusted with PBST.	10
7	Supernatant is incubated with antibody-coupled MNPs.	30
8	Complex of flagellin-antibody-coupled MNPs is separated and eluted in PBST.	20
9	SPR assay	10
	Total Time	200

**Table 2 sensors-22-00475-t002:** Comparisons of performance of the three SPR assays.

Direct Assay	Sequential Two-Step Sandwich Assay	Preincubation One-Step Sandwich Assay
Less sample preparation (total time 150 min)	Less sample preparation (total time 150 min)	Additional incubation (total time 200 min)
One-step injection	Two-step injection	One-step injection
Signal from flagellin only (no amplification)	Separated signals from flagellin and MNPs (amplification)	Signal from the complex of flagellin and MNPs (amplification)
Ratio of amplification to flagellin signal = 1	Ratio of amplification to flagellin signal = 7.5	Ratio of amplification to flagellin signal = 14

## Data Availability

The data presented in this study are available on request from the authors.

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
