# Peer review of "Magnetic Nanoparticles Enhanced Surface Plasmon Resonance Biosensor for Rapid Detection of Salmonella Typhimurium in Romaine Lettuce"

_sensors, 2022, doi:10.3390/s22020475_

Round 1
Reviewer 1 Report
In this manuscript” Magnetic Nanoparticles Enhanced Surface Plasmon Resonance Biosensor for Rapid Detection of Salmonella Typhimurium in Romaine Lettuce”, the authors report the development of a rapid and sensitive S. Typhimurium detection method using a surface plasmon resonance (SPR) biosensor combined with the amplification of antibody coupled MNPs. Although different methods are available to timely detect Salmonella in foods, SPR has benefit of real-time detection with high sensitivity and specificity. Therefore, the authors have developed a functional protocol to detect S. Typhimurium starting from romaine lettuce sample preparation to SPR detection. The proposed detection methods are interesting and can provide a way for the potential of SPR biosensor in combination with MNPs for the detection of other pathogens and biomolecules with higher sensitivity and specificity. I suggest that the manuscript is suitable for publication provided that the following minor issues being addressed.
- There are many format details in the manuscript that should be revised. For example, the clarity of the diagram is not uniform. The font of Fig. 2a is blurred and the font of Fig. 2b is clear. In addition, Figure 4 is labeled “ABCD”, but Figure 5 is not labeled “ABC”. The picture sizes of the “ABCD” in Figure 4 are inconsistent, such as “Time” size. The top three pictures in Figure 5 are not aligned.
- The theoretical part and design idea of the whole manuscript are very clear. But I think it will be clearer to express the characteristics through tables. For the convenience of readers, I would suggest the authors give a table to summarize the advantages, disadvantages and characteristics of each method.
- This work employed nanostructured surface plasmon resonance for sensing applications. However, the introduction part only involves sensing aspect, the background about nanostructure and surface plasmon resonance are seldom mentioned, such as Materials 11(11), 2315 (2018),Journal of Materials Science 54(8), 6301-6309 (2019).
Author Response
1. There are many format details in the manuscript that should be revised. For example, the clarity of the diagram is not uniform. The font of Fig. 2a is blurred and the font of Fig. 2b is clear. In addition, Figure 4 is labeled “ABCD”, but Figure 5 is not labeled “ABC”. The picture sizes of the “ABCD” in Figure 4 are inconsistent, such as “Time” size. The top three pictures in Figure 5 are not aligned.
Response 1: The Fig. 2a and 2b have been adjusted to improve clarity. Picture sizes of “ABCD” in Figure 4 have been adjusted proportionally. Figure 5 has been labeled with “ABC” and pictures “ABC” have been aligned.
2. The theoretical part and design idea of the whole manuscript are very clear. But I think it will be clearer to express the characteristics through tables. For the convenience of readers, I would suggest the authors give a table to summarize the advantages, disadvantages and characteristics of each method.
Response 2: A new table (Table 2) is added for the comparisons of performance of the three SPR assays.
3. This work employed nanostructured surface plasmon resonance for sensing applications. However, the introduction part only involves sensing aspect, the background about nanostructure and surface plasmon resonance are seldom mentioned, such as Materials 11(11), 2315 (2018), Journal of Materials Science 54(8), 6301-6309 (2019).
Response 3: A description of the nanoparticle structure is added in the Introduction.
Reviewer 2 Report
Report on “Magnetic Nanoparticles Enhanced Surface Plasmon Resonance Biosensor for Rapid Detection of Salmonella Typhimurium in Romaine Lettuce”
This is a quite good article that might be suitable for publication after some clarifications that will certainly improve this manuscript.
- Figure 1. It would be great if there was a scale and then it would be possible to represent the real dimensions of the objects.
- Figure 3 looks unfortunate as it contains a lot of unused space.
- The same for Fig.4 and 5.
- Line 333. Appearing here Fe3O4 seems incomprehensible, unexpected and paradoxical. Perhaps it makes sense to devote several sentences to this oxide in introduction. Why interesting, what goals, how is it grown, recent achievements. Please, give more motivational explanations why Fe3O4? Why Fe3O4 and not Fe2O3. More information about size and preparation methods will be useful as well as their possible influence on results. For a wider range of readers, more information on Fe3O4 preparation methods would be desirable. See for examples, some of them published this year in MDPI journals,
Serga, V.et al . Impact of Gadolinium on the Structure and Magnetic Properties of Nanocrystalline Powders of Iron Oxides Produced by the Extraction-Pyrolytic Method. Materials 2020, 13, 4147.
Li, Y.; et al. Superparamagnetic α-Fe2O3/Fe3O4 Heterogeneous Nanoparticles with Enhanced Biocompatibility. Nanomaterials 2021, 11, 834
Kahil, H., Faramawy, A., El-Sayed, H., & Abdel-Sattar, A. (2021). Magnetic properties and SAR for gadolinium-doped iron oxide nanoparticles prepared by hydrothermal method. Crystals, 11(10), 1153.
- Finally, analysis of the bibliography says that the relevance of the work has not been demonstrated. Most of the references are over 15 years old. And therefore, it is unclear to what extent this topic is still of interest.
Author Response
1. Figure 1. It would be great if there was a scale and then it would be possible to represent the real dimensions of the objects.
Response 1: Figure 1 has been revised to reflect the proportions of the objects.
A description of the properties and size of the MNP is added in the Introduction.
2. Figure 3 looks unfortunate as it contains a lot of unused space.
Response 2: Figure 3 has been formatted to reduce the unused space.
3. The same for Fig.4 and 5.
Response 3: Figure 4 and 5 have been formatted to reduce the unused space.
4. Line 333. Appearing here Fe3O4 seems incomprehensible, unexpected and paradoxical. Perhaps it makes sense to devote several sentences to this oxide in introduction. Why interesting, what goals, how is it grown, recent achievements. Please, give more motivational explanations why Fe3O4? Why Fe3O4 and not Fe2O3. More information about size and preparation methods will be useful as well as their possible influence on results. For a wider range of readers, more information on Fe3O4 preparation methods would be desirable. See for examples, some of them published this year in MDPI journals,
Serga, V.et al . Impact of Gadolinium on the Structure and Magnetic Properties of Nanocrystalline Powders of Iron Oxides Produced by the Extraction-Pyrolytic Method. Materials 2020, 13, 4147.
Li, Y.; et al. Superparamagnetic α-Fe2O3/Fe3O4 Heterogeneous Nanoparticles with Enhanced Biocompatibility. Nanomaterials 2021, 11, 834
Kahil, H., Faramawy, A., El-Sayed, H., & Abdel-Sattar, A. (2021). Magnetic properties and SAR for gadolinium-doped iron oxide nanoparticles prepared by hydrothermal method. Crystals, 11(10), 1153.
Response 4: The sentence at Line 333 has been revised to avoid the confusion.
5. Finally, analysis of the bibliography says that the relevance of the work has not been demonstrated. Most of the references are over 15 years old. And therefore, it is unclear to what extent this topic is still of interest.
Response 5: This study is aimed to develop a sample preparation and SPR amplification protocol for the detection of Salmonella. This study is relevant to the advance of two previous works published in 2019.
Bhandari, D.; Chen, F.-C.; Hamal, S.; Bridgman, R. C. Kinetic analysis and epitope mapping of monoclonal antibodies to Salmonella Typhimurium flagellin using a surface plasmon resonance biosensor. Antibodies 2019, 8, 22.
Bhandari, D.; Chen, F.-C.; Bridgman, R. C. Detection of Salmonella Typhimurium in romaine lettuce using a surface plasmon resonance biosensor. Biosensors 2019, 9, 94.